# DNA Damage Response during Replication Correlates with CIN70 Score and Determines Survival in HNSCC Patients

**DOI:** 10.3390/cancers13061194

**Published:** 2021-03-10

**Authors:** Ioan T. Bold, Ann-Kathrin Specht, Conrad F. Droste, Alexandra Zielinski, Felix Meyer, Till S. Clauditz, Adrian Münscher, Stefan Werner, Kai Rothkamm, Cordula Petersen, Kerstin Borgmann

**Affiliations:** 1Laboratory of Radiobiology & Experimental Radiooncology, Center of Oncology, University Medical Center Hamburg-Eppendorf, 20246 Hamburg, Germany; theodor.bold@gmail.com (I.T.B.); tutu22303@gmail.com (A.-K.S.); a.zielinski@uke.de (A.Z.); fe.meyer@uke.de (F.M.); k.rothkamm@uke.de (K.R.); 2University Cancer Center Hamburg (UCCH), University Medical Center Hamburg-Eppendorf, 20246 Hamburg, Germany; cdroste@mpi-bremen.de; 3Department of Pathology, University Medical Center Hamburg-Eppendorf, 20246 Hamburg, Germany; t.clauditz@uke.de; 4Department of Otorhinolaryngology and Head and Neck Surgery, University Medical Center Hamburg-Eppendorf, 20246 Hamburg, Germany; a.muenscher@uke.de; 5Department of Tumorbiology, University Medical Center Hamburg-Eppendorf, 20246 Hamburg, Germany; st.werner@uke.de; 6Department of Radiotherapy and Radiation Oncology, University Medical Center Hamburg-Eppendorf, 20246 Hamburg, Germany; cor.petersen@uke.de

**Keywords:** chromosomal instability (CIN), CIN70 score, replication stress, Chk1 inhibition, HNSCC, radiotherapy, radiosensitization

## Abstract

**Simple Summary:**

Functional aneuploidy as determined by expression of the 70 genes of the CIN70 score, influences prognosis. The importance of DNA repair protein expression and activity in this context is unclear. We can show here that a high CIN70 score is associated with increased expression of 44 proteins of the known DNA repair complexes. Among these, an association with survival was only observed for 12 proteins of DNA replication and replication-associated DNA repair. This suggests that it is not the expression of individual DNA repair proteins of a DNA repair complex that causes resistance to therapy, but rather a balanced expression and coordinated activation of corresponding signaling cascades. Inhibitors to generally block the S-phase DNA damage response should therefore be used to develop therapeutic strategies in the future.

**Abstract:**

Aneuploidy is a consequence of chromosomal instability (CIN) that affects prognosis. Gene expression levels associated with aneuploidy provide insight into the molecular mechanisms underlying CIN. Based on the gene signature whose expression was consistent with functional aneuploidy, the CIN70 score was established. We observed an association of CIN70 score and survival in 519 HNSCC patients in the TCGA dataset; the 15% patients with the lowest CIN70 score showed better survival (*p* = 0.11), but association was statistically non-significant. This correlated with the expression of 39 proteins of the major repair complexes. A positive association with survival was observed for MSH2, XRCC1, MRE11A, BRCA1, BRCA2, LIG1, DNA2, POLD1, MCM2, RAD54B, claspin, a negative for ERCC1, all related with replication. We hypothesized that expression of these factors leads to protection of replication through efficient repair and determines survival and resistance to therapy. Protein expression differences in HNSCC cell lines did not correlate with cellular sensitivity after treatment. Rather, it was observed that the stability of the DNA replication fork determined resistance, which was dependent on the ATR/CHK1-mediated S-phase signaling cascade. This suggests that it is not the expression of individual DNA repair proteins that causes therapy resistance, but rather a balanced expression and coordinated activation of corresponding signaling cascades.

## 1. Introduction

Head and neck squamous cell carcinoma (HNSCC) is the sixth most common cancer and accounts for approximately 4–6% of all newly diagnosed tumors. Advanced HNSCC has a poor prognosis and survival rates of approximately 50% have not changed significantly in recent years. Current treatment regimens combining radiotherapy with standard chemotherapeutic agents such as cisplatin, 5-fluorouracil, and etoposide have been shown to be effective; however, treatment response is determined by the intrinsic genetics of the tumor, and the development of appropriately tailored therapeutic options is desirable. Genomic features are therefore promising to potentially provide biomarkers of treatment response. Previous studies have identified frequently mutated genes through extensive DNA sequencing and genetic analysis, revealing so-called driver genes in HNSCC [1], the prognostic value of which is the subject of ongoing studies [2]. Mutations in DNA repair genes were observed with striking frequency in HNSCC. This particularly affects genes of mismatch repair (MMR), the DNA double-strand break repair pathway homologous recombination (HR), and DNA cross-link repair (ICL repair) [3]. These mutations lead to alterations in protein expression, also seen in HNSCC cell lines [4], and negatively impact survival [5]. While the DNA damage response provides cellular drug and radiation resistance through repair [4], it also controls c stability. Defects in DNA repair thus promote genomic instability [6] and allow cancer cells to acquire new genetic traits, including those necessary to become resistant to therapy. Indeed, the degree of genomic instability or DNA repair status of a tumor is associated with poor prognosis in cancer [7,8]. Measurements of genomic instability and its surrogates such as DNA repair defects vary. Reliable markers of this could be obtained by genomic profiling. In addition, chromosomal instability (CIN) is observed in tumors, resulting in random irregularities in the distribution of chromosomes during mitosis due to defective DNA repair and DNA replication, but also defective segregation in mitosis. Thus, chromosomal instability (CIN) describes gains and losses of whole chromosomes (aneuploidy) or parts of chromosomes (insertions or deletions) [9,10], and also offers itself as a potential marker. Beside genomic and chromosomal instability, the CIN70 score has been established [11]. This uses a signature of genes associated with functional segmental aneuploidy. Based on this signature of genes, negative prognostic impact of CIN70 score on survival was observed for several tumor entities [11]. Important questions regarding the prognostic value of the CIN70 score, along with the expression of other DNA repair genes in HNSCC, are pending.

Most studies on prognosis and resistance to therapy in HNSCC have focused exclusively on expression analyses of individual DNA repair genes without considering functional analyses of the affected DNA repair complexes. Furthermore, it was not specified which expression profiles could be used to develop alternative therapeutic approaches. In the search for suitable alternatives, considerable recent interest has focused on DNA repair pathways as potential targets for novel cancer treatments [12]. The poly(adenosine diphosphate ribose) polymerase (PARP) family of DNA end-binding core proteins is involved in one such DNA repair pathway [13], and inhibition of PARP1 (PARPi) has emerged as a promising therapeutic approach for the treatment of certain cancers, particularly breast and ovarian cancers. We have also shown for HNSCC cell lines that HR-deficient cell lines can be radiosensitized by PARP1 inhibition [4]. We and others further observed that PARP1, in addition to its direct involvement in repair processes in HR-deficient HNSCC cell lines, also plays a role in controlling replication [14,15]. Other potential inhibitors of candidate genes with a function in replication, such as CHK1 and ATR [16], have been the subject of preclinical and clinical investigation in combination with irradiation and also showed promise for specific radiosensitization of genomically unstable HNSCC tumor cell lines [4,17]. Therefore, our goal is to analyze the prognostic significance of the CIN70 score in HNSCC and the impact of DNA repair protein expression using the open-access Cancer Genome Atlas (TCGA) dataset and to further characterize the impact of the identified DNA repair processes after DNA damage by ionizing and UV irradiation, Mitomycin C and topotecan using preclinical models.

## 2. Results

### 2.1. Functional Aneuploidy (CIN70) Inconclusively Determines Poorer Survival in HNSCC Patients

To investigate the relationship, of functional aneuploidy and tumoral mRNA expression of DNA repair proteins for survival of HNSCC patients, the CIN70 score was first determined in 519 tumor samples of the TCGA dataset ([11]; Figure 1A). We observed a trend toward better survival for the group of 15% of patients with the lowest CIN70 score compared to the rest of patients, which was not statistically significant (*p* = 0.11).

Functional aneuploidy is not only influenced by the expression of genes considered in the CIN70 score but is critically dependent on the functionality of the DNA repair machinery. The CIN70 score does not include DNA repair proteins other than MLH6, FEN1 and RAD51AP1 [6]. Therefore, the mRNA expression of 44 key proteins of the major DNA repair pathways for the 15% patients with low CIN70 score was compared to those with high CIN70 score (Figure 1B–H). Here, the general DNA repair damage response mediated by the ATR/ATM signaling cascade (Figure 1B), mismatch repair (MMR; Figure 1C), single-strand break repair (SSBR; Figure 1D), the two DNA repair pathways of DNA double-strand break repair, Homologous recombination (HR) (Figure 1E) and classical non-homologous end-joining (cNHEJ) (Figure 1F), interstrand cross-link repair (ICL repair; Figure 1G), and proteins with a function in stabilizing the DNA replication fork (Figure 1H) were considered. RPA1, EXO1, LIG1, LIG3, BLM, and MRE11A are not exclusive to a single DNA repair pathway and were therefore shown multiple times. Variation in housekeeping genes was shown in blue bars. 39 of the 44 DNA repair proteins analyzed showed mostly highly significant increased mRNA expression in samples with a high CIN70 score compared to samples with a low CIN70 score; only ATM showed a nonsignificant increase at a high CIN70 score. In contrast, significantly lower mRNA expression in tumor samples with high CIN70 score was observed for SMC1 (**), ERCC1 (***) and LIG4 (****), and a non-significant for XRCC4 (n.s.) and BOD1L1 (n.s.). Thus, only the DNA repair complexes MMR, HR and ICL repair showed a general upregulation of mRNA expression in tumors with high CIN70 score (Figure 1C,E,G). This led to the question of whether there could be a continuous increase in mRNA expression with the increase in CIN70 score. To clarify this question, expression was plotted against the CIN70 score for each protein. Figure 2A (top) exemplifies for RAD54B and BRCA1 the clearest continuous increase in expression with CIN70 score, (r^2^ = 0.5049; *p* = 0.0001) and (r^2^ = 0.4228; *p* = 0.0001) respectively. A significant correlation was also observed for RAD51 expression with r^2^ = 0.5299; *p* < 0.0001. In contrast, for MRE11A and ERCC1 (Figure 2A, bottom), there was no relationship of mRNA expression and CIN70 score, with r^2^ = 0.016; *p* = 0.004 and r^2^ = 0.0008; n.s. All other DNA repair proteins analyzed showed a lower relationship of expression and CIN70 score compared with RAD51 and BRCA1, but a higher one than MRE11A and ERRC1 (data not shown). Thus, there appears to be a continuous correlation of mRNA expression and genomic instability, particularly for the HR proteins.

### 2.2. DNA Repair Protein Expression Does Not Reflect Cellular Resistance

As previously shown by us and others, high protein expression of HR proteins negatively affected patient survival in several tumor entities [18]. This implies that also in HNSCC high protein expression of HR proteins could lead to resistance to treatment through increased repair capacity. For HNSCC cell lines, we showed in vitro that HR capacity varied up to 30-fold (Figure 2B; [4]). However, this variation was not reflected by differences in the protein expression of BRCA1, ATR, MRE11A, CHK1, and RAD51 as well as BLM, Exo1, DNA2 and PARP1 in these cell lines (Figure 2C and Appendix A, and [4]. Cell lines with pronounced expression of RAD51 or BRCA1 showed low HR capacity and vice versa. This suggests that cells lines with low HR capacity had a high RAD51 expression (Figure 2B and Wurster et al., 2016). Therefore, sensitivity to mitomycin C, another independent marker of potential HR deficiency, was investigated. It appeared that a wide variation in sensitivity existed, which did not correlate with HR deficiency. Cell lines with the lowest HR capacity showed both, the highest resistance or sensitivity. After irradiation (Figure 2D middle), the same pattern was seen; HR-incompetent cell lines showed both low and high radiation resistance.

These marked differences in resistance were also evident for treatment after topotecan (Figure 2D right). Surprisingly, the MMC- and IR-resistant cell lines showed higher sensitivity here. In conclusion, differences in HR capacity do not seem to be conclusive in terms of genomic instability and resistance to tumor therapy. Rather, it appears that resistance may be attributable to neither an oversupply nor an undersupply of DNA repair protein, but rather to a balanced composition of proteins required within a DNA repair complex.

### 2.3. Expression of Replication-Associated Proteins Impacts Survival of HNSCC Patients

To test the hypothesis of whether increased or low expression of DNA repair proteins compared to a balanced supply of DNA repair proteins affects survival of HNSCC patients, the 15% extreme values for each DNA repair protein based on highest and/or lowest expression were pooled and compared to the rest of the patients (Figure 3). Expression was found to be associated with patient survival for 12 of the 44 proteins analyzed (Figure 3), with nine proteins showing significance and three showing no significance. 11 of the 12 proteins showed a positive and only ERCC1 (*p* = 0.029) showed a negative association with survival by comparing the 15% high/low expression versus the remaining 85% of patients. Strikingly, neither proteins of the ATR/ATM signaling cascade nor the NHEJ had a positive or negative prognostic effect on survival (data not shown). In contrast, one protein each of MMR or SSBR showed better patient survival at low/high expression, significant for MSH2 with *p* = 0.00095 and non-significant for XRCC1 with *p* = 0.087 (Figure 3, top). Remarkably, three proteins directly attributable to the HR pathway, BRCA1 (*p* = 0.036), BRCA2 (*p* = 0.022), RAD54B (*p* = 0.0054) (Figure 3, top center) and two other proteins required for HR, such as LIG1 (*p* = 0.025) and MRE11A (*p* = 0.091), positively influenced survival at low/high expression, with all except MRE11A showing significance (Figure 3, bottom center). The other proteins affecting HNSCC patient survival were CLSPN (*p* = 0.061), DNA2 (*p* = 0.0014), POLD1 (*p* = 0.046) and MCM2 (*p* = 0.038), whose functions contribute to the stability and error-free function of the replication machinery (Figure 3, bottom).

### 2.4. Resistance Is Mediated by Replication Fork Stability and Active DNA Damage Response of S Phase

To test the hypothesis that therapy resistance manifests during S-phase, DNA replication processes in the two most resistant and most sensitive cell lines were studied in detail [19,20,21,22]. The focus was on both the protection of the already replicated DNA strand (fork protection) and the restart of replication after damage repair (fork recovery) (Figure 4A,B). Halting the replication machinery by depletion of nucleotides (HU treatment) showed no degradation of the already synthesized DNA strand for any of the four cell lines examined (Appendix A). However, after irradiation, it was apparent that the two radiation-sensitive cell lines FaDu and XF354 (Figure 2D) showed a significant shortening of the DNA strand already synthesized before irradiation, whereas the two resistant cell lines showed no difference compared with unirradiated cells (Figure 4C).

Thus, radiation-sensitive cell lines appear to have significantly reduced protection of already synthesized DNA compared to radiation-resistant cell lines after damage induction. Next, the effect of irradiation on the capability to restart replication processes after damage repair was examined (Figure 4D). Again, the two radiation-sensitive cell lines showed significantly delayed recovery of replication compared to the two resistant cell lines, with approximately 50% and 25% shorter DNA fibers, respectively. In contrast, the radiation-resistant cell lines showed almost no impairment of replication by irradiation, visible by the slight significant shortening (HSC4) or slight non-significant lengthening (SAS) of DNA fibers. Here, the observed ability to protect replication forks is directly related to the observed radiosensitivity. A significant correlation of fork protection with cellular survival (Figure 2D) was observed after irradiation with 6 Gy for seven cell lines examined, with r^2^ = 0.62; *p* = 0.04 (Figure 4E).

Replication is mainly controlled by activation of the kinase CHK1 through phosphorylation [19,20,21]. To study CHK1 activation specifically in S phase, UV irradiation was used [22] (Figure 4F). Already the endogenous CHK1 level showed differences between the cell lines, with a slightly lower expression of CHK1 in the sensitives. After UV irradiation, all cell lines examined showed a time-dependent activation of CHK1 up to 60 min after irradiation, expressed as the rate of pCH1 to CHK1 (Figure 4F bottom). However, the activation of CHK1 was increased 4-fold in the resistant compared to the sensitive cell lines. To verify these activation differences of CHK1, DNA replication processes were examined in one resistant and one sensitive HNSCC cell line after UV irradiation (Figure 4G).

This showed that the radiation-sensitive cell line FaDu exhibited a significant 50% shortening of the already synthesized DNA strand, whereas the resistant cell lines showed no difference in comparison to non-irradiated cells (Figure 4G left). In contrast, there were no differences in replication fork recovery, visible by a threefold significant shortening of DNA fibers after UV irradiation in both cell lines (Figure 4G right). Thus, the strong activation of CHK1 seems to be directed in particular to a protection of the already synthesized DNA strand. To confirm this hypothesis, CHK1 was inhibited in one radioresistant and one radiosensitive cell line and the subsequent effects on DNA replication processes and cellular survival were investigated (Figure 5). There was a concentration-dependent decrease in CHK1 expression with increasing siRNA concentration, with an almost complete reduction at a concentration of 150 nm siRNA compared to scrRNA in both cell lines (Figure 5A). A concentration of 150 nm was used for further experiments. First, the effects of CHK1 reduction on replication processes were investigated (Figure 5B). For the radiosensitive cell line, there was no further reduction in replication tract length of the strand synthesized before irradiation with 6 Gy after siRNA against CHK1 compared with untreated cells and cells treated with scrRNA (Figure 5B). In contrast, the radiation-resistant cell line showed a significant reduction of the DNA strand already synthesized before irradiation, exclusively when CHK1 was inhibited (Figure 5C). To confirm this observation, the effect of the CHK1 inhibitor MK6778 on replication processes was also examined. Again, the radiation-resistant showed a highly significant shortening of the strand replicated before irradiation, while FaDu cells showed only a minor shortening (Figure 5B,C). Next, the effect of inhibition of CHK1 on cellular survival after irradiation was analyzed in each of two radiation-sensitive and resistant cell lines (Figure 5D). There was a clear, significant sensitization of the two radioresistant cell lines. The two radiosensitive cell lines showed only a slight, non-significant radiosensitization by CHK1 inhibition. Thus, these data confirm the assumption that the observed radioresistance is due to enhanced activation of CHK1.

## 3. Discussion

### 3.1. CIN70 Score in HNSCC

In this study, CIN70 signature did not conclusively stratify HNSCC tumors according to clinical outcome of HNSCC patients. This was evident when the 15% patients with the lowest CIN70 score were stratified against the rest of the patients. A further decrease in the impact of CIN70 score on survival was seen when the lower quartile was contrasted against the rest of the patients (data not shown) [9]. Other studies focused on the impact of genomic alterations and survival of HNSCC patients. Thus, by analyzing copy number variation using CGH, a significant association with survival in oropharyngeal squamous cell carcinomas OOSCC (*p* = 0.003) was observed when the patient cohort was stratified into low, high and very high level genomic alterations [23]. In contrast, no significant association was observed in OSCC when copy number variation of chromosomes 1 and 4 was analyzed via FISH [24]. A significant association of point mutations in CCND1 and CDKN2A, as well as chromosomal instability based on copy number variation, was associated with progression-free survival in oropharyngeal and hypopharyngeal carcinoma independent of HPV status according to mutation analyses of 556 genes [25]. Mutations in TP53, Notch1, and KDR were identified as prognostic independent markers by next generation sequencing in HNSCC [26]. Mutations in TP53, CCND1, CDKN2A, and FGFR1 genes also showed association with HNSCC survival after analysis of targeted next generation sequencing in HNSCC [2].

Summarizing the data, genomic and chromosomal instability, and in particular the expression of genes included in the CIN70-score, seem to be poor predictors of survival in HNSCC patients. One of the possible reasons for this could be the great heterogeneity of HNSCC, which already differ significantly when comparing subgroups in terms of survival. This could be a cause for the low association of CIN70 score and survival of this study, as the TCGA dataset includes relatively high numbers of oral cavity carcinomas. Furthermore, HPV status, especially in OOSCC also significantly influences survival. Thus, analyses of this type should consider classification into HPV+ and HPV- tumors [27]. It is also possible that it is not the sum of mutations or the sum of genes affected by mutations [28], but the expression of genes that prevent mutations and DNA damage is of greater importance. Because the CIN70 score only involves the DNA repair proteins MLH6, RAD51AP1, and FEN1 [11], the interest in this study was the significance of differences in expressed DNA repair proteins.

Here, we show that HNSCC tumors with an elevated CIN70 score have significantly increased expression of almost all DNA repair proteins (Figure 1B–H), even in DNA repair complexes expressed in a proliferation-independent manner (Figure 1B,F). In fact, a simultaneous increase was only seen for a few proteins, often those of the HR complex, whereas most of the analyzed proteins were expressed independently of the CIN70 score (Figure 2A). This was also reflected by our in vitro experiments, where an increased expression of HR-proteins did not automatically lead to increased HR capacity [4] and showed both resistance and sensitivity according to the respective damage setting (Figure 2B–D). This suggests that DNA repair counteracts functional aneuploidy and influences therapy resistance only when the concentration of required key proteins is balanced. Survival analyses confirm this hypothesis, showing the highest survival in a combined analysis of the 15% extreme values (Figure 3). This confirms observations by [29] who previously described this as a paradox of the importance of genomic instability for survival. They hypothesized that genomic instability improves the biological fitness of cancer cells from a life-sustaining level to a threshold level. [29]. Outside this range, cell viability would be reduced and a better response to therapy would be given. It is possible that the functionality of DNA repair processes, which is determined by the interplay of up to 50 proteins, is only effective when ideally balanced and thus has a negative effect on therapy response. This could also explain that overexpression of Ku70/80 and RAD51, respectively, both positively and negatively impacts HNSCC survival [18,30,31,32,33]. For ERRC1, XRCC1, MMR proteins, and XPF, a lower loco-regional control, shorter progression-free survival, and negative outcome have always been observed with elevated expression in prior HNSCC studies [34,35,36,37]. It is possible that consideration of low expression would further emphasize the impact on patient survival. Alternatively, focusing on processes that directly trigger chromosomal instability, such as mitosis and replication, could narrow down appropriate candidate genes.

### 3.2. Replication Fork Stability Determines Resistance in HNSCC

One of the possible sources of genomic instability are defective DNA replication processes. Protection of the active replication fork is the top priority for maintaining genomic stability. This is ensured by the presence of all active DNA repair complexes [38]. Here, the proteins of HR have a direct responsibility for protecting the replication fork in addition to their predominant role in DNA repair by attaching to open DNA strands. In the absence of RAD51, BRCA1, BRCA2, and FANCD2, degradation of already synthesized DNA occurred already after depletion of the nucleotide pool or inhibition of the replicative polymerase, which impeded replication fork progression but did not cause direct DNA damage. This degradation can be compensatorily prevented by overexpression of RAD51 [39,40]. Nucleolytic degradation of MRE11A is primarily thought to be responsible for the degradation [41,42] and later studies showed similar effects also for DNA2, Mus81 or Exo1 [43,44].

The HNSCC cell lines we studied did not show nucleolytic degradation after nucleotide depletion (Appendix A). Surprisingly, the radiosensitive cell lines showed several micrometers of degradation of the DNA synthesized before irradiation (Figure 4C). Such degradation of the pre-synthesized DNA strand was also observed after acetaldehyde treatment and UV irradiation [45,46]. The feasible reasons for such degradation would be that it is an initiated DNA repair process or pure nucleolytic degradation of unprotected DNA by MRE11A or other endonucleases. An already initiated DNA repair process (fork regression) is rather unlikely, since the probability that 6 Gy irradiation directly affects active replication forks physiologically is statistically unlikely. This suggestion is confirmed by the observation that DNA strand degradation is maintained when only the exchange activity, i.e., the domain necessary for DNA repair, of RAD51 is mutated, but the protein retains the ability to attach to single-stranded DNA [43]. Rather, the function of CHK1 in preventing degradation appears to be of paramount importance here, (a) through direct association with the DNA replication fork [15], (b) control of dormant replication origins, mediated by ATR activation of the intra-s phase checkpoint [20], (c) control of the nuclease activity of Mus81 [47]. Together, our selective radiosensitizing effect of CHK1 inhibition of radioresistant HNSCC cells (Figure 5) and the observation that most proteins informative of HNSCC patient survival were directly or indirectly associated with replication (Figure 3), suggest that inhibitors to generally inactivate the S-phase DNA damage response, such as CHK1 or ATR [16], are promising options for future advancement of existing therapies.

## 4. Materials and Methods

### 4.1. Clinical in Silico Analysis

Clinical and mRNA expression data were extracted from the TCGA database of the cBioportal (http://www.cbioportal.org). For each tumor, the CIN70 score was calculated according to [11] by including the expression values of CIN70 genes from 519 patients. For the calculation of survival probability, the 15% of patients with the lowest expression of the CIN70 genes (CIN70 low) were compared with the remaining patients. For analysis of MSH2, XRCC1, ERCC1, BRCA1, BRCA2, RAD54B, Lig1, CLSPN, MRE11A, DNA2, POLD2, or MCM2 protein mRNA expression, the 15% patients with the lowest or highest expression were also grouped against the remaining patients and analyzed using a log-rank test.

### 4.2. Cell Culture and Treatments

Primary squamous carcinoma cell lines (University of Turku-Squamous cell carcinoma (UT-SCC)) have been established previously in the Grenman lab, Turku University, Finland and FaDu, Cal33, SAS, HSC4 and XF354 were from ATCC/LGC (Wesel, Germany). Cells were cultivated in DMEM supplemented with 10% FCS, 2 mM glutamine, 100 U/mL penicillin and 100 μg/mL streptomycin at 37 °C at 10% CO_2_. For the determination of cellular survival cells were treated with up to 1.5 μg/mL to mitomycin C (Medac, Wedel, Germany) or 1 µM topotecan (Sigma, St. Louis, MS, USA,) for 6 h or 24 h and irradiation experiments were performed using radiation doses up to 6 Gy at a dose rate of 1.2 Gy/min using a Gulmay X-ray machine (GULMAY Medical, Byfleet, UK) or UV 5mJ (BioRad Laboratories, Berkeley, CA, USA). All treatments were performed in 37 °C and 10% CO_2_ atmosphere. CHK1 inhibition was achieved by using siGENOME human CHEK1 siRNA (Horizon Discovery, Cambridge, UK) up to 150 nM for 24 h or the small molecule inhibitor MK8776 (Selleck Chemicals, Houston, TX, USA) at 2 µM for 2 h.

### 4.3. Western Blot

Total protein was extracted from exponentially growing cells at passage 8–10 and 40 g/mL were resolved by SDS-PAGE using a 4–15% gradient gel (Bio-Rad Laboratories). After transfer and blocking overnight at 4 °C in Odyssey Blocking Buffer (Li-Cor, Lincoln, NE, USA) proteins were detected by primary antibodies against BRCA1 [2A-9] (1:500, kindly provided by Stephen Smith, Leibnitz Institute, Jena, Germany), ATR [N-19] (Santa Cruz, St. Cruz, CA, USA, 1:1000), CHK1 [2G1D5] (Cell Signaling, Danvers, MA, USA, 1:750), RAD51 [14B4] (1:2.000, GeneTex, Irvine, CA, USA), MRE11A [12D7] (Abcam, Cambridge, UK, 1:500), pCHK1 [Ser296] (Cell Signaling, 1:1000), -actin [AC-74] (1:50.000, Sigma, St. Louis, MO, USA). Primary antibodies were detected with IRDYE 680 conjugated anti-mouse IgG, IRDYE 800 conjugated anti-rabbit IgG (Li-Cor, 1:7500), IRDYE 680 conjugated anti-rabbit IgG (Licor, 1:7.500 or 15.000) or IRDYE 800 conjugated anti-mouse IgG (Li-Cor 1:7.500 or 15.000). Quantitative and qualitative analysis was done by using Li-Cor Odyssey (Li-Cor, Lincoln, NE, USA).

### 4.4. DNA Fiber Assay

Exponentially growing cells were pulse labeled with 25 µM CldU (Sigma) followed by 250 µM IdU (Sigma) for 45 min each. Cells were irradiated with ionizing or UV between both labels, CHK1 inhibitor was given 2 h before labelling. Labeled cells were harvested, DNA fiber spreads prepared and stained as described [19]. Fibers were examined using an Axioplan 2 fluorescence microscope (Zeiss, Oberkochen, Germany). CldU and IdU tracks were measured using ImageJ (version 1.48, Wayne Rasband, NIH, USA) [19,20,21,22,48]. At least 300 forks were analyzed.

### 4.5. Clonogenic Survival

For colony formation assay 250 cells were seeded in a 6-well plate 6 h before treatment and cells were cultured for 14 days. Cells were fixed and stained with 1% crystal violet (Sigma-Aldrich, St. Louis, MO, USA). Colonies with more than 50 cells were determined microscopically and normalized to untreated samples. Each survival curve represents the mean of at least three independent experiments.

### 4.6. Patient Survival and Statistical Analysis

Patient survival analyses were performed according to the Kaplan–Meier method and the Log-rank test using the R packages “survival” and “survminer” [49]. Statistical analysis, curve fitting and graphs were performed using Prism 9.02 (GraphPad Software, San Diego, CA, USA) or SPSS, IBM, Chicago, IL, USA). Data are given as mean (+SEM) of 3–5 replicate experiments. Significance was tested by log-rank or Student’s *t*-test.

## 5. Conclusions

In the era of precision medicine, tumor-specific DNA damage defects are increasingly emerging as attractive targets. With the successful introduction of PARP1 inhibitors in clinical trials, multiple opportunities have emerged. CHK1 and ATR inhibitors may represent a further contribution to even more successful treatment strategies. The baseline situation for all these therapies is the complete loss of corresponding DNA repair proteins. It remains unclear what consequences a limiting reduction or increase in the expression of individual proteins has on the interplay and function of an entire DNA repair complex and how this affects the response of existing and new therapies. This should be the subject of further investigations in order to make the application of these promising therapeutic approaches accessible to an even larger patient population.

## Figures and Tables

**Figure 1 cancers-13-01194-f001:**
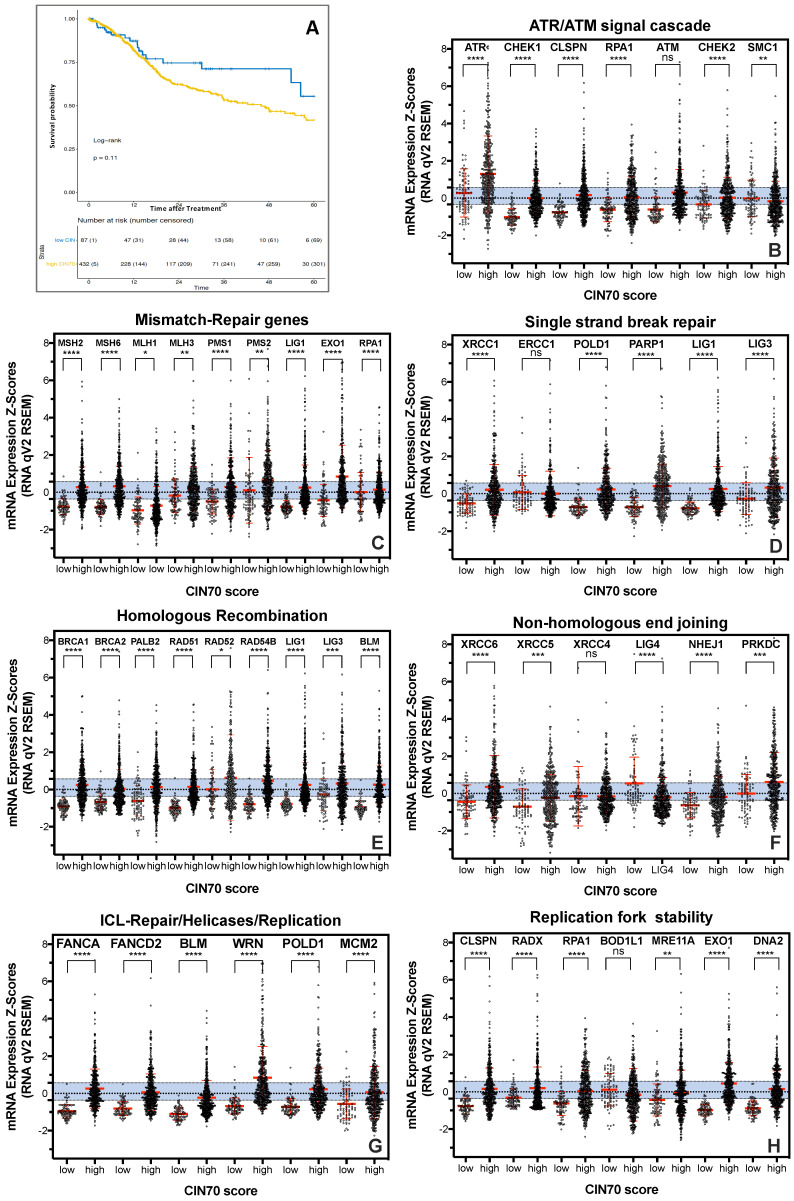
Functional aneuploidy indicates worse survival in HNSCC patients and is associated with high mRNA expression of DNA repair proteins. (**A**) Probability of survival after treatment of tumors with a low compared to a high CIN70 score; (**B**–**H**) mRNA expression in tumors with low and high CIN70 scores involved in the ATR/ATM signaling cascade, mismatch repair, single-strand break repair, homologous recombination, classical non-homologous end-joining, ICL repair and DNA replication fork stability (h) was analyzed. Asterisks (*) represent significant differences (* *p* < 0.05; ** *p* < 0.01; *** *p* < 0.001; **** *p* < 0.0001; Student´s *t*-test). Blue bars represent variation of housekeeping genes.

**Figure 2 cancers-13-01194-f002:**
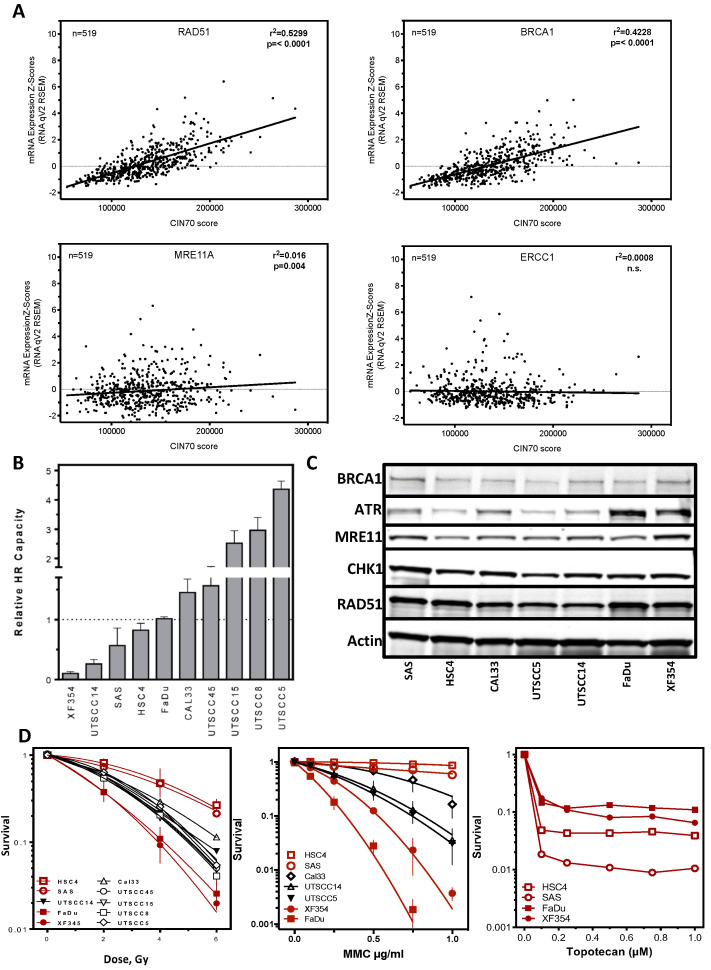
mRNA expression of HR proteins correlates with CIN70 score (**A**) The mRNA expression for the proteins RAD54B, BRCA1, MRE11, and ERCC1 was plotted against the corresponding CIN70 score of 519 HNSCC patients and calculated with linear regression. (**B**) Relative HR capacity in HNSCC cell lines (data taken from Wurster et al., 2016). (**C**) Protein expression of BRCA1, ATR, MRE11A, CHK1, and RAD51 of 7 unirradiated HNSCC cell lines. (**D**) Cellular survival after irradiation, mitomycin C and topotecan determined by the colony formation assay.

**Figure 3 cancers-13-01194-f003:**
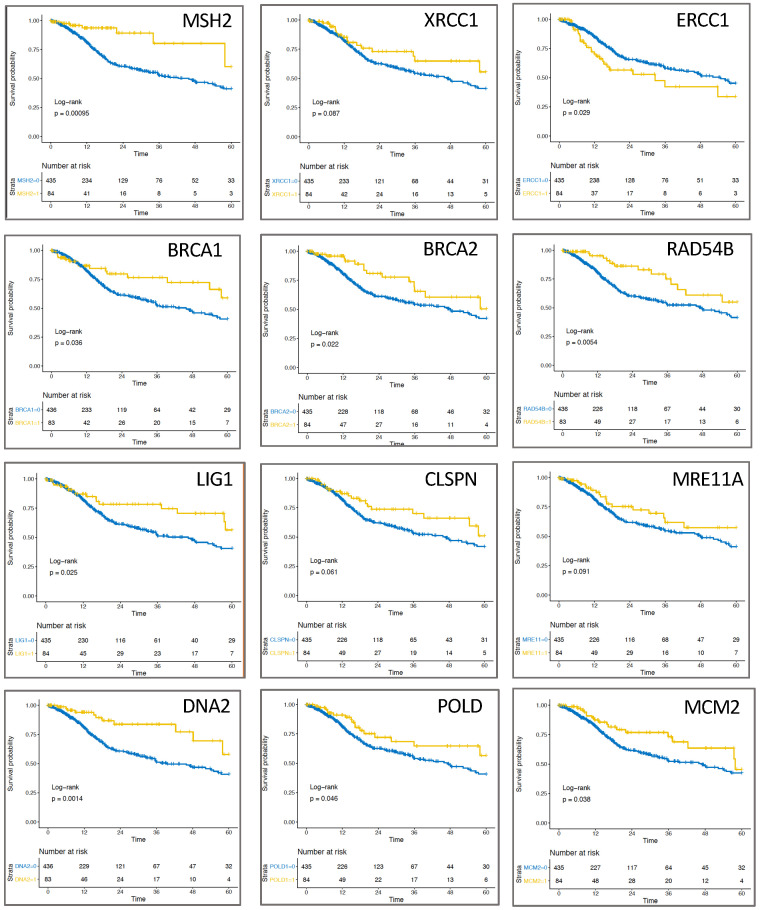
Survival probability is determined by mRNA expression of the DNA repair proteins MSH2, XRCC1, ERCC1, BRCA1, BRCA2, RAD54B, LIG1, CLSPN, MRE11A, DNA2, POLD1 and MCM2 in 519 HNSCC patients of the TCGA dataset using Kaplan-Meier survival curves up to 60 months after therapy. The 15% of patients with the lowest and/or highest protein expression were compared with the remaining patients. Differences between both groups were determined by the log-rank test.

**Figure 4 cancers-13-01194-f004:**
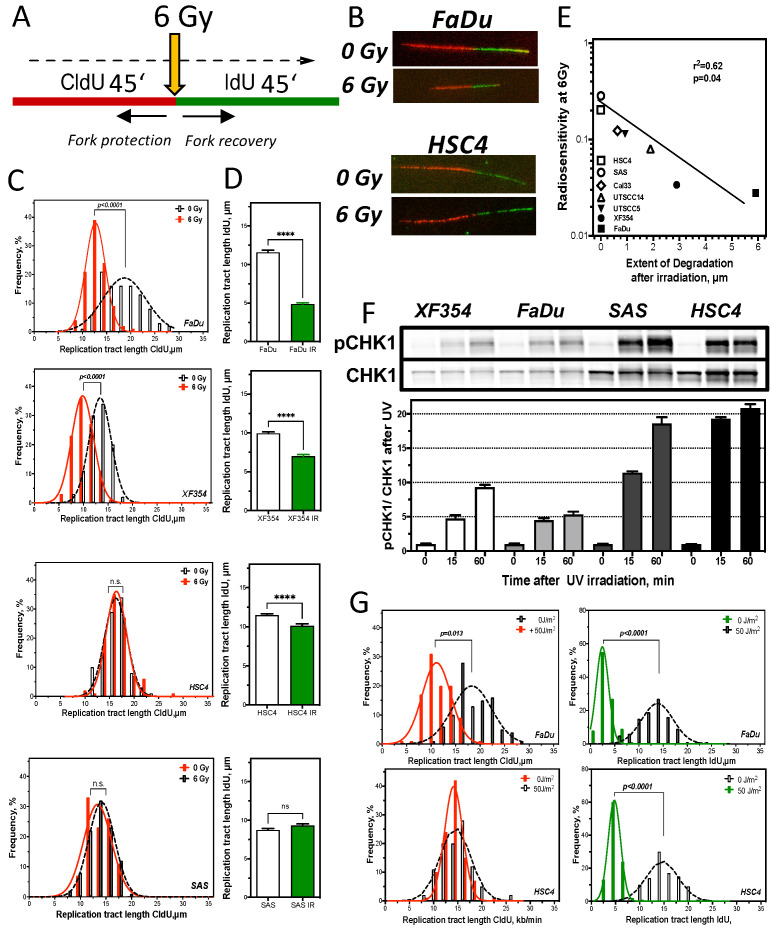
Resistance is mediated by DNA replication fork stability and active DNA damage response of S phase. (**A**,**B**) Treatment scheme and examples of replication tracts and replication tract lengths before (CldU) (**C**) and after (IdU) (**D**) irradiation. Correlation of replication tract length and cellular radiosensitivity (**E**). (**F**) Qualitative and quantitative activation of CHK1/pCHK1 after UV irradiation. Replication tract length before (left) and after UV irradiation (right) (**G**). *p*-values represent errors of the mean. Asterisks (*) represent significant differences (**** *p* < 0.0001; Student´s *t*-test).

**Figure 5 cancers-13-01194-f005:**
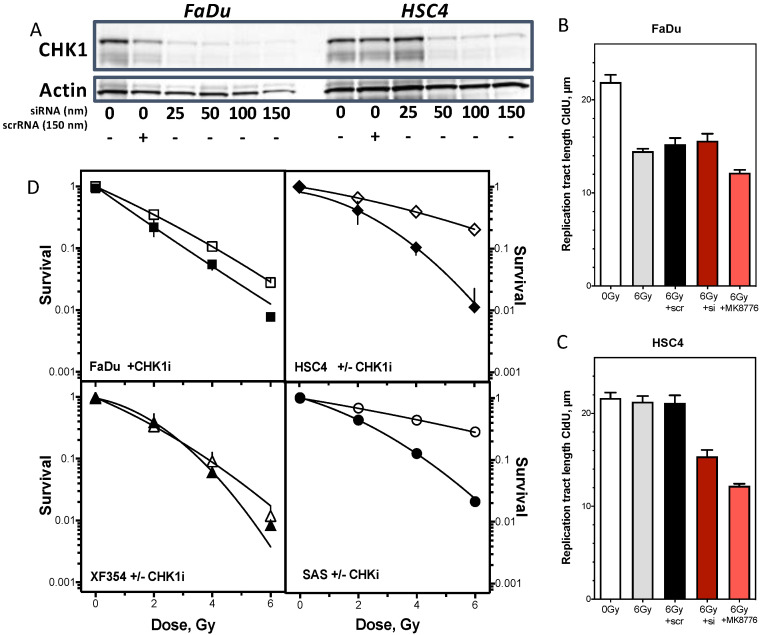
Inhibition of S phase checkpoint control radiosensitizes exclusively resistant HNSCC cell lines. (**A**) Concentration-dependent inhibition of CHK1 by siRNA treatment in FaDu and HSC4 cells. (**B**,**C**) Effect of siRNA or CHK1 inhibitor MK8776 treatment in combination with irradiation on replication tract length compared to irradiation alone. (**D**) Cellular radiosensitization only in resistant cell lines. Error bars represent the error of means of at least three independent experiments. (Student´s *t*-test).

## Data Availability

The datasets analyzed and novel reagents used during the current study are available from the corresponding author upon request and after material transfer agreement.

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
