# Peer review of "DNA Damage Response during Replication Correlates with CIN70 Score and Determines Survival in HNSCC Patients"

_cancers, 2021, doi:10.3390/cancers13061194_

Round 1
Reviewer 1 Report
In this article, the authors investigated chromosomal instability (CIN), as determined by the expression of the 70 genes of the CIN70 score, and its impact on response to cancer therapy. The authors analyzed data about HNSCC patients and performed experiments using radiosensitive and radioresistant cell lines.
Overall it is a well-described manuscript, discussion part is supported by many figures and data, but a few parts need clear explanation. Therefore, here I’m presenting my suggestions and questions divided into reviewed parts of the article.
Simple summary and abstract parts
- In the sentence: “inhibitors for general inactivation of the S-phase DNA damage response…” – please consider instead of the word “inactivation” another word like blocking, etc. which is more popular in biological terminology of cell-cycle.
- Authors are using the term “tumor therapy” – what kind of tumor therapy do you mean? We know that different types of signaling pathways are observed between radiotherapy, chemotherapy, or immunotherapy, please precise.
Introduction
- You are using the term “TCGA dataset” in the introduction part – please explain what does it mean, I know that authors put information in the Materials and methods part about that, but is it a free available dataset or not?
- Also as a goal of that paper, please add that your intention is studying effects after therapy, or before or in combined.
Materials and Methods
- Still, I am not sure what means low CIN70 score – please elaborate
- You are using doses up to 6 Gy – it’s a relatively high dose as a single fraction, please explain why this dose was chosen.
- It is known that different doses of radiation resulted in different gene expression patterns. Do you suppose that the same effects will be seen after 2 or 4 Gy of X-rays, please explain how different doses of radiation and their energy can influence observed effects?
- In a sentence: “Total protein was extracted from exponentially growing cells” – please add a passage of each cell line was used. It’s known that cell’s response may differ between stage of cell lines and stage of the cell cycle as well.
- Please explain in detail why UV was used. I found only that statement: “To study CHK1 activation specifically in S phase, UV irradiation was used” - please cite a reference confirming that process.
Results
- Figure 2 is not visible.
- Figure 4 description – please explain how replication tracts and replication tract lengths before (C) and after (D) irradiation were obtained on this Figure.
The discussion should be elaborated presenting limitations of this study and future perspectives.
Author Response
Reviewer 1
In this article, the authors investigated chromosomal instability (CIN), as determined by the expression of the 70 genes of the CIN70 score, and its impact on response to cancer therapy. The authors analyzed data about HNSCC patients and performed experiments using radiosensitive and radioresistant cell lines.
Overall it is a well-described manuscript, discussion part is supported by many figures and data, but a few parts need clear explanation. Therefore, here I’m presenting my suggestions and questions divided into reviewed parts of the article.
Simple summary and abstract parts
- In the sentence: “inhibitors for general inactivation of the S-phase DNA damage response…” – please consider instead of the word “inactivation” another word like blocking, etc. which is more popular in biological terminology of cell-cycle.
We thank the reviewer for this constructive suggestion and have replaced inactivation by blocking in the revised manuscript
- Authors are using the term “tumor therapy” – what kind of tumor therapy do you mean? We know that different types of signaling pathways are observed between radiotherapy, chemotherapy, or immunotherapy, please precise.
We thank the reviewer for the understandable objection and have replaced tumortherapy by radiochemotherapy
Introduction
- You are using the term “TCGA dataset” in the introduction part – please explain what does it mean, I know that authors put information in the Materials and methods part about that, but is it a free available dataset or not?
- Also as a goal of that paper, please add that your intention is studying effects after therapy, or before or in combined.
We thank the reviewer for this comment and have amended the introduction in line with the suggestions as follows:
Therefore, our goal is to analyze the prognostic significance of the CIN70 score in HNSCC and the impact of DNA repair protein expression using the open-access Cancer Genome Atlas (TCGA) dataset and to further characterize the impact of the identified DNA repair processes after DNA damage by ionizing and UV irradiation, Mitomycin C and topotecan using preclinical models.
Materials and Methods
- Still, I am not sure what means low CIN70 score – please elaborate
For better comprehensibility, we have changed the corresponding sentence in the Material & Methods section as follows: For the calculation of survival probability, the 15% of patients with the lowest expression of the CIN70 genes (CIN70 low) were compared with the remaining patients.
- You are using doses up to 6 Gy – it’s a relatively high dose as a single fraction, please explain why this dose was chosen.
Irradiation leads to a linear quadratic decrease in survival, i.e., at low doses the shoulders of the curves overlap, and it looks like the cell lines being compared have no difference in radiosensitivity. This only becomes apparent when higher doses are used. Therefore, we used doses of 0-6 Gy for the colony formation assays and in later experiments a dose at which we could assume that the differences in radiosensitivity were manifest. 2. irradiation induces about 3000 DNA damages per Gy; this is relatively low considering the sum of opened replication forks. Therefore, we also selected a higher dose at which we could be sure that opened replication forks would be perturbed by the damage.
- It is known that different doses of radiation resulted in different gene expression patterns. Do you suppose that the same effects will be seen after 2 or 4 Gy of X-rays, please explain how different doses of radiation and their energy can influence observed effects?
We agree with the reviewer that the dose level used may affect protein expression. We therefore only looked at the expression of unirradiated cells, according to the TCGA dataset. We have adapted the legend of Figure 2 accordingly.
Only after UV irradiation, we checked the activation of CHK1 directly, 15 and 30 minutes after irradiation by phosphorylation.
- In a sentence: “Total protein was extracted from exponentially growing cells” – please add a passage of each cell line was used. It’s known that cell’s response may differ between stage of cell lines and stage of the cell cycle as well.
We thank the reviewer for this comment and have added the missing information as follows: Total protein was extracted from exponentially growing cells at passage 8-10 and 40 g/ml were resolved by SDS-PAGE using a 4%-15% gradient gel.
- Please explain in detail why UV was used. I found only that statement: “To study CHK1 activation specifically in S phase, UV irradiation was used” - please cite a reference confirming that process.
We have added an explanatory publication in the revised version of the manuskript
Results
- Figure 2 is not visible.
We apologize for this. We have reinserted the figure and hope that it is visible in the revision.
- Figure 4 description – please explain how replication tracts and replication tract lengths before (C) and after (D) irradiation were obtained on this Figure.
We have now added the following to the figure legend:
(A, B) Treatment scheme and examples of replication tract lengths and replication tract lengths before (CldU) (C) and after (IdU) (D) irradiation.
The discussion should be elaborated presenting limitations of this study and future perspectives.
We thank the reviewer for this constructive proposal and have now provided a possible outlook under point 5 Conclusion
Reviewer 2 Report
In this study, authors analysed the prognostic significance of the CIN70 score in HNSCC and the impact of DNA repair protein expression of 44 proteins, using the TCGA dataset and preclinical models. Results were clearly presented with detailed description focused on all relevant information to answer the questions raised, including questions raised throughout the analysis. Authors showed a negative impact of CIN70 score on survival of HNSCC patients, with a good approach comparing the 15% patients with the lowest CIN70 score with the rest of the patients. Since, the manuscript is well written, experiments well designed and results and discussions well presented according to the goal described in the introduction, I recommend the publication of the manuscript in the present form.
In the lines 149 and 151, please close the parentheses.
Author Response
Reviewer 2
In this study, authors analysed the prognostic significance of the CIN70 score in HNSCC and the impact of DNA repair protein expression of 44 proteins, using the TCGA dataset and preclinical models. Results were clearly presented with detailed description focused on all relevant information to answer the questions raised, including questions raised throughout the analysis. Authors showed a negative impact of CIN70 score on survival of HNSCC patients, with a good approach comparing the 15% patients with the lowest CIN70 score with the rest of the patients. Since, the manuscript is well written, experiments well designed and results and discussions well presented according to the goal described in the introduction, I recommend the publication of the manuscript in the present form.
In the lines 149 and 151, please close the parentheses.
Thank you for the positive review of our manuscript.
We have closed the parenthesis in line 149
Reviewer 3 Report
In the manuscript (MS) “DNA damage response during replication correlates with CIN70 score and determines survival in HNSCC patients” Bold et al. analyzed the prognostic significance of the CIN70 score in HNSCC and the impact of DNA repair protein expression using the TCGA dataset and further characterized the impact of the identified DNA repair processes using preclinical models.
- The authors reported a correlation of CIN70 score and survival in 519 HNSCC patients in the TCGA dataset: “There was a trend toward better survival when the group 15% patients with the lowest CIN70 score were compared with the rest of the patients (p=0.11)” – Results, L115-116. But p value 0.11 (log-rank test) shows that association between low CIN score and better survival is not statistically significant, and to interpret this as a trend toward better survival is an exaggeration. Moreover, there is a statement in the Abstract: “We observed a correlation of CIN70 score and survival in 519 HNSCC patients in the TCGA dataset; the 15% patients with the lowest CIN70 score showed better survival” (L36-37) which is not supported by the data. Again, similar misleading statement in Discussion (L242-244): "In this study, we showed for the first time a negative impact of CIN70 score on survival of HNSCC patients. This was visible when stratifying the 15% patients with the lowest CIN70 score toward the rest of the patients"
- Why patients were grouped based on 15% of lowest CIN70 score vs 85% of the remaining patients? What happened if threshold would be 20% vs 80%? or 25% vs 75%?
- mRNA expression of 44 DNA repair proteins was compared in the groups of patients with the high (85%) and low (15%) CIN70 score (note, that some of the repair proteins are also part of CIN70 set). In Results: “39 out of 44 DNA repair proteins analyzed showed mostly highly significant increased mRNA expression in samples with a high CIN70 score compared to samples with a low CIN70 score; only ATM (n.s.), MLH1(*), MLH3 (**), PMS2 (**) LIG1 (*), RAD52 (*), and MRE11A (**) showed a nonsignificant increase at a high CIN70 score.” – L127-131. At the same time, corresponding Figure 1 B-H shows that nonsignificant increase was observed for ATM, ERCC1, RXCC4, and BOD1L1, but not for MLH1, MLH3, PMS2, LIG1, RAD52 and MRE11A.
- A positive association with survival was reported for 12 proteins of DNA replication and replication-associated DNA repair (L29-30, Simple Summary). In the Abstract there are only 10 of them... In the Results (L119-120): “12 of the 44 proteins analyzed showed a significant effect on patient survival. 11 proteins showed a positive effect and one (ERCC1; p=0.029) a negative effect on patient survival when expressed at high/low levels.” But what does it mean: a significant effect on patient survival? Is it relevant to statistical significance? Figure 3 shows that among those 12 proteins only 9 have p value < 0.05 for survival probability, and tree of them (XRCC1, MRE11A, CLSPN) do not show statistically significant differences.
These are just few examples of inaccuracies and discrepancies which are present in the MS.
Another big problem of this MS is the frequent inappropriate use of the term “chromosomal instability”, “CIN” instead of “aneuploidy”, and “genomic instability” instead of “genomic imbalances”. The CIN70 score derived by Carter et al. (2006) cannot be used as a surrogate measurement of chromosomal instability but rather as a proxy for the overall level of aneuploidy assessed at the transcriptional level. In the original publication by Carter et al., abbreviation CIN70 was applied to a signature of genes that correlated with functional segmental aneuploidy (clonal segmental genomic imbalances), and not with chromosomal instability. To call it CIN70 was misleading and is the source of confusion among researchers who used this 70 gene score in their studies. The terms “aneuploidy” and “chromosomal instability” are not interchangeable. CIN and aneuploidy are different traits and they are likely to have distinct impacts on clinical tumor behavior. Chromosomal instability (CIN) is a process of creation of random chromosomal aberrations, non-clonal structural and numerical chromosomal changes, and is measured by the rates of instability (Geigl et al., 2008; Lengauer et al., 1998; van Jaarsveld, Kops, 2016; Yuen, 2010; Duijf, Benezra, 2013; etc…). Term “aneuploidy” is usually used to describe the presence of genomic imbalances, clonal deviations from the euploid chromosomal complement. Aneuploidy is a state, not a process. The rate of occurrence of random chromosomal aberrations (CIN) may not correlate with the level of aneuploidy because many other factors are involved in clonal selection of chromosomal aberrations leading to aneuploidy.
Conclusion: The MS requires a substantial rewriting with correction of inaccuracies and discrepancies as well as clarification of used terminology.
Author Response
In the manuscript (MS) “DNA damage response during replication correlates with CIN70 score and determines survival in HNSCC patients” Bold et al. analyzed the prognostic significance of the CIN70 score in HNSCC and the impact of DNA repair protein expression using the TCGA dataset and further characterized the impact of the identified DNA repair processes using preclinical models.
- The authors reported a correlation of CIN70 score and survival in 519 HNSCC patients in the TCGA dataset: “There was a trend toward better survival when the group 15% patients with the lowest CIN70 score were compared with the rest of the patients (p=0.11)” – Results, L115-116. But p value 0.11 (log-rank test) shows that association between low CIN score and better survival is not statistically significant, and to interpret this as a trend toward better survival is an exaggeration. Moreover, there is a statement in the Abstract: “We observed a correlation of CIN70 score and survival in 519 HNSCC patients in the TCGA dataset; the 15% patients with the lowest CIN70 score showed better survival” (L36-37) which is not supported by the data. Again, similar misleading statement in Discussion (L242-244): "In this study, we showed for the first time a negative impact of CIN70 score on survival of HNSCC patients. This was visible when stratifying the 15% patients with the lowest CIN70 score toward the rest of the patients"
We thank the reviewer for pointing this out and have added: non-significant in all the listed examples.
- Why patients were grouped based on 15% of lowest CIN70 score vs 85% of the remaining patients? What happened if threshold would be 20% vs 80%? or 25% vs 75%?
In fact, we also tested the top quartile against the bottom quartile, the bottom 25% against the rest, and observed only trends without significance. This motivated us to perform further analyses because we suspected that in addition to the DNA repair genes listed in the CIN70 score, RAD51AP1, FEN1, and MSH6, which are important in breast cancer, glioblastoma, and NSCLC, other DNA repair proteins may also be significant in HNSCC survival, as shown in Figure 3.
- mRNA expression of 44 DNA repair proteins was compared in the groups of patients with the high (85%) and low (15%) CIN70 score (note, that some of the repair proteins are also part of CIN70 set). In Results: “39 out of 44 DNA repair proteins analyzed showed mostly highly significant increased mRNA expression in samples with a high CIN70 score compared to samples with a low CIN70 score; only ATM (n.s.), MLH1(*), MLH3 (**), PMS2 (**) LIG1 (*), RAD52 (*), and MRE11A (**) showed a nonsignificant increase at a high CIN70 score.” – L127-131. At the same time, corresponding Figure 1 B-H shows that nonsignificant increase was observed for ATM, ERCC1, RXCC4, and BOD1L1, but not for MLH1, MLH3, PMS2, LIG1, RAD52 and MRE11A.
We apologize for the inaccuracies and have changed numbers and significances in the text according to the reviewer's suggestion.
- A positive association with survival was reported for 12 proteins of DNA replication and replication-associated DNA repair (L29-30, Simple Summary). In the Abstract there are only 10 of them... In the Results (L119-120): “12 of the 44 proteins analyzed showed a significant effect on patient survival. 11 proteins showed a positive effect and one (ERCC1; p=0.029) a negative effect on patient survival when expressed at high/low levels.” But what does it mean: a significant effect on patient survival? Is it relevant to statistical significance? Figure 3 shows that among those 12 proteins only 9 have p value < 0.05 for survival probability, and tree of them (XRCC1, MRE11A, CLSPN) do not show statistically significant differences.
We apologize for the inaccuracies and have changed the text in the revision as follows:
Expression was found to be associated with prognosis for 12 of the 44 proteins analyzed (Figure 3), with 9 proteins showing a significant trend and 3 showing a non-significant trend. 11 of the 12 proteins showed a positive and only ERCC1 (p=0.029) showed a negative association with prognosis when comparing the 15% high/low expression versus the remaining 85% of patients. Strikingly, neither one protein of the ATR/ATM signaling cascade nor the NHEJ had a positive or negative prognostic effect on survival (data not shown). In contrast, one protein each of MMR or SSBR showed better patient survival at low/high expression, significant for MSH2 with p=0.00095 and non-significant for XRCC1 with p=0.087 (Figure 3, top). Remarkably, three proteins directly attributable to the HR pathway, BRCA1 (p=0.036), BRCA2 (p=0.022), RAD54B (p=0.0054) (Figure 3, top center) and two other proteins required for HR, such as LIG1 (p=0.025) and MRE11A (p=0.091), positively influenced survival at low/high expression (Figure 3, bottom center). The other positive prognostic proteins were CLSPN (p=0.061), DNA2 (p=0.0014), POLD1 (p=0.046) and MCM2 (p=0.038), whose functions contribute to the stability and error-free function of the replication machinery (Figure 5, bottom).
These are just few examples of inaccuracies and discrepancies which are present in the MS.
Another big problem of this MS is the frequent inappropriate use of the term “chromosomal instability”, “CIN” instead of “aneuploidy”, and “genomic instability” instead of “genomic imbalances”. The CIN70 score derived by Carter et al. (2006) cannot be used as a surrogate measurement of chromosomal instability but rather as a proxy for the overall level of aneuploidy assessed at the transcriptional level. In the original publication by Carter et al., abbreviation CIN70 was applied to a signature of genes that correlated with functional segmental aneuploidy (clonal segmental genomic imbalances), and not with chromosomal instability. To call it CIN70 was misleading and is the source of confusion among researchers who used this 70 gene score in their studies. The terms “aneuploidy” and “chromosomal instability” are not interchangeable. CIN and aneuploidy are different traits and they are likely to have distinct impacts on clinical tumor behavior. Chromosomal instability (CIN) is a process of creation of random chromosomal aberrations, non-clonal structural and numerical chromosomal changes, and is measured by the rates of instability (Geigl et al., 2008; Lengauer et al., 1998; van Jaarsveld, Kops, 2016; Yuen, 2010; Duijf, Benezra, 2013; etc…). Term “aneuploidy” is usually used to describe the presence of genomic imbalances, clonal deviations from the euploid chromosomal complement. Aneuploidy is a state, not a process. The rate of occurrence of random chromosomal aberrations (CIN) may not correlate with the level of aneuploidy because many other factors are involved in clonal selection of chromosomal aberrations leading to aneuploidy.
Conclusion: The MS requires a substantial rewriting with correction of inaccuracies and discrepancies as well as clarification of used terminology.
We thank the author for his comments regarding the differences of genomic and chromosomal instability and the specificity of the CIN70score we used, which is functional aneuploidy. We have revised the language of the entire manuscript in this regard and hope to have removed the inaccuracies.

Round 2
Reviewer 3 Report
The new version of the MS contains several changes, mainly cosmetic. They are partial and do not improve the MS enough to be considered acceptable for publication in its current format. Many misleading, inaccurate or exaggerated statements are still there. Most importantly, all conclusions should be clearly supported by the results. This MS requires deep re-thinking and re-writing. Summary, Abstract, Introduction, Results, Figures (1-3) and Discussion - every section of the MS requires revision, especially in the sections relevant to CIN70 score and gene expression in the patients' samples.
There is another part of the MS (almost unrelated to the first one) that describes experiments with HNSCC cell lines. This part is well written and clearly presented, with an adequate interpretations of the obtained results. The MS would greatly benefit if the same high standards were applied to the whole MS.
Author Response
Dear reviewer,
we have generally revised the manuscript with regard to your comments, changed and adapted text passages, included new literature and have the impression that it has indeed become significantly better due to your points of criticism. Many thanks for this.

Round 3
Reviewer 3 Report
There are very substantial improvements in the MS, especially, in the Introduction and Discussion.
In the other parts of the MS some discrepancies are still present.
1) Abstract, L29-30: "We show here that a high CIN70 score is associated with increased expression of 44 proteins of the known DNA repair complexes."
This disagrees with the results:
Results, L125-129: 39 of the 44 DNA repair proteins analyzed showed mostly highly significant increased mRNA expression in samples with a high CIN70 score compared to samples with a low CIN70 score; only ATM showed a nonsignificant increase at a high CIN70 score. In contrast, significantly lower mRNA expression in tumor samples with high CIN70 score was observed for SMC1 (**), ERCC1 (****) and LIG4 (****), and a non-significant for XRCC4 (n.s.) and BOD1L1 (n.s.).
2) In Abstract, L40-42: "We observed an association of CIN70 score and survival in 519 HNSCC patients in the TCGA dataset; the 15% patients with the lowest CIN70 score showed better survival (p=0.11)" - association was statistically non-significant.
In Results, Ln 107: “ Functional aneuploidy determines poorer survival in HNSCC patients” – actually, this should be: Functional aneuploidy (CIN70 score) inconclusively determines poorer survival in HNSCC patients
Ln 108-112: “To investigate the relationship, of functional aneuploidy and tumoral mRNA expression of DNA repair proteins for survival of HNSCC patients, the CIN70 score was first determined in 519 tumor samples of the TCGA dataset (Carter et al., 2006; Figure 1A). We observed better survival for the group of 15% of patients with the lowest CIN70 score compared to the rest of patients, which was not statistically significant (p=0.11).” – how better survival was observed if there was no statically significant difference in the survival of the group of 15% of patients with the lowest CIN70 score compared to the rest of patients?
Figure 1. “Functional aneuploidy determines worse survival in HNSCC patients and is associated with high mRNA expression of DNA repair proteins.” – misleading Figure legend.
Discussion: "In this study, we observed a negative non-significant effect of CIN70 score on survival of HNSCC patients. .........the expression of genes included in the CIN70-score seem to be poor predictors of survival in HNSCC patients." - correct, in this study CIN70 signature did not conclusively stratify HNSCC tumors according to clinical outcome. Please make it clear also in the Abstract and in Results section.
3) "Figure 3. Survival probability is determined by mRNA expression of the DNA repair proteins MSH2, XRCC1, ERCC1, BRCA1, BRCA2, RAD54B, LIG1, CLSPN, MRE11A, DNA2, POLD1 and MCM2 in 519 HNSCC patients of the TCGA dataset using Kaplan-Meier survival curves up to 60 months after therapy. The 15% of patients with the lowest CIN70 score were compared with the remaining patients. Differences between both groups were determined by the log-rank test."
L163-166: "To test the hypothesis of whether increased or low expression of DNA repair proteins compared to a balanced supply of DNA repair proteins affects survival of HNSCC patients, the 15% extreme values for each DNA repair protein based on highest and/or lowest expression were pooled and compared to the rest of the patients (Figure 3)."
How 15% of tumors were selected in this experiment? Based on CIN70 score or based on highest/lowest expression of DNA repair proteins?
4) L137-138: "All other DNA repair proteins shown here are within the correlations (data not shown)." - what does that mean?
5) Ln378-380: "It is still unclear what significance a possibly limiting reduction or increased expression of individual proteins has on the interactions with individual proteins or whole DNA repair complexes and the response to existing and novel therapies." - needs to be rewritten more clearly.
After additional corrections this MS can be published.
Author Response
Dear reviewer,
We have taken into account your last suggested changes for the revision of the manuscript and hope that all reservations for a publication could be removed with it. All changes made can be seen in the attached Word-file. We would be pleased if the manuscript could be published in its present form and thank you for improving the manuscript and your patience,
Yours sincerely, Kerstin Borgmann
